# Factors Associated with the Use of Industrial Liquid Silicone among Travesti and Transgender Women in Salvador, Northeast Brazil

**Ricardo Araújo da Silva [1,2,\*]**, **Luís Augusto Vasconcelos da Silva [1,3]**, **Fabiane Soares [1]** and **Inês Dourado [1]**

1    Instituto de Saúde Coletiva, Universidade Federal da Bahia, Salvador 40110-040, Brazil
2    Escola Bahiana de Medicina e Saúde Pública, Av. Silveira Martins, Salvador 40290-000, Brazil
3    Instituto de Humanidades, Artes e Ciências, Universidade Federal da Bahia, Salvador 40170-115, Brazil
\*    Correspondence: ricardosilva@bahiana.edu.br; Tel.: +55-(71)-991744425

**Abstract:** Background: The illicit use of industrial liquid silicone (ILS) is a common practice among travesti and transgender Women (TrTW) in the process of bodily change. The "pumped ladies" apply the ILS without any preparation regarding biosafety, and this practice poses serious risks to the health of TrTW, including death. This study aims to describe the prevalence of ILS use and estimate the associated factors among TrTW in a Brazilian city. Methods: This behavioral and serological survey employed the Respondent-Driven Sampling (RDS) recruitment technique. Participants who declared themselves to be travesti or transgender women over 15 years were considered eligible. The sample consisted of 127 TrTW. The bivariate analysis estimated the prevalence of ILS use by sociodemographic, cultural, and behavioral variables. Multivariate analysis used Poisson regression for adjusted estimates with respective 95% confidence intervals. Results: Approximately 31.6% of the TrTW (n = 44) used ILS and 57.2% (n = 67) were under 25 years old. Most (73.2%, n = 79) had access to more than eight years of schooling. Nine percent (n = 15) tested positive for HIV, and 31.6% (n = 48) tested positive for syphilis. The TrTW who felt more comfortable with their body image had a threefold higher prevalence of ILS use and those who performed an HIV test before the study had a 4.5 times higher prevalence of ILS. Conclusion: ILS is widely used by TrTW in Brazil. Although public policies for the transgenderizing process exist, this process still occurs illicitly.

**Keywords:** industrial liquid silicone; travesti; transgender women





## 1. Introduction

In Brazil, the terms travesti and transgender are used by the transgender community itself. The differential marker between these identities is the political activism or the subjectivity of travesti and transgender women (TrTW) [1]. However, these identity distinctions/borders are liquid, unstable, blurred, and in movement and interaction [2]. When reflecting on trans people's access to health services, Santos [3] uses the term "trans" not to homogenize or simplify the gender experience, but to highlight that trans people value social trajectories, the body change experience, the daily violence endured, and the hardships they experience in order to subvert the hegemonic gender and sexuality model. (1). However, there is no simple definition that may characterize such populations to fit the requirements of both theoretical issues and their translation into concrete studies carried out in the field. In Brazil, and in some Latin American countries, the terms "travestis", "transsexual women", or "transgender women" are commonly used by individuals and communities themselves.

Inequities among TrTW can be explained by contexts of vulnerability, for instance, structural, including poor socioeconomic conditions (e.g., when gender identity represents a barrier to ensuring stable employment) and difficult access to prevention and care services (e.g., discrimination in health services by health professionals and a lack of qualified

care); interpersonal vulnerability, such as stigma, discrimination, and violence driven by gender identity (transphobia) within social interactions which creates further vulnerabilities; and the individual and biological vulnerability dimensions created by sexual behaviors and practices (more money offered for higher risk sexual acts such as unprotected sex, condomless anal sex, number of partners, or illicit substance use). Furthermore, TrTW face several social, economic, and access barriers to healthcare, which lead them to risk exposures, such as the injectable use of industrial liquid silicone (ILS). Many inject ILS in order to modify physical traits, seeking female performance and the desired body [4,5]. Body change also occurs with hormone use associated with other technologies, such as hair prostheses, hair removal, and tattoos. Larissa Pelúcio [4] emphasizes that the travesti body change process occurs in stages as if rituals were being followed in endless rites. Thus, the travesti transformed body she calls "travesti being" is a process with no closure and which requires constant care [6]. This process witnesses the desire to be seen socially as a cisgender woman, treated as such in public spaces, or in terms of access to health services [7].

Without specialized professional care, the body change resources used by TrTW put their health at risk. The rites of these bodily changes, experienced clandestinely, are permeated with risks and sequelae without proper biosafety [3]. After World War II, ILS use started in the 1940s from toxicology publications whose content indicated ILS as "physiologically inert". The interest of the medical community in biocompatible materials has grown since then. This product, used for shaping body contours, became popular in Germany, Switzerland, and Japan. However, in 1965, new studies indicated that large amounts of SLI injected into the body could cause irreversible damage [8].

Illicitly used ILS is a common practice among TrTW for body modification in low- and middle-income countries. Travesti injecting ILS into other TrTW are known as "pumped ladies"—this term usually refers to older and more experienced travesti who have acquired the practice of injecting silicone or applying hormones [9]. This usually entails the application of silicone to the buttocks, face, and breasts [10,11], without biosafety criteria, which poses serious health risks to TrTW, including risks ranging from tissue necrosis, embolization, artery obstructions, pain, and migration of injected material, which can lead to death [4,12]. The TrTW are submitted to the application of this product unsafely in order to modify their bodies [4] although they are aware of the risk.

A study in a hospital in Paris [13] found that all 77 TrTW using ILS had dermatological complications, including lymphatic or subcutaneous migration of the ILS, inflammation, varicose veins, post-inflammatory pigmentation, infection, and abscesses. A qualitative study conducted in Rio de Janeiro evidenced that ILS use also caused thrombosis and infections, mainly in the lower limbs, requiring hospitalization [14].

This study describes the prevalence of ILS use and estimates the associated factors among TrTW in Salvador, Northeast Brazil.

## 2. Material and Methods

This investigation is nested within the PopTrans study, a behavioral and serological survey carried out in the capital city of Salvador, Northeast Brazil, with TrMT living in the city or its metropolitan region for at least three months and aged 15 years or over [15], from 2014 to 2016. Participants were recruited through the Respondent-Driven Sampling (RDS) or participant-directed sampling method.

### 2.1. Seed Sampling and Recruitment

RDS is a chain-link sampling method that begins with "seeds"—a convenience sample of members of the target population chosen by the researchers [16]. Previous qualitative formative research was conducted in order to know the population, the places of sociability, and the participants who would launch the recruitment process. At this stage of the research, focus groups and in-depth interviews were carried out with TrMT recruited from social movements and places of sociability. Initial participants were selected as seeds,

after the formative research, in an attempt to better assess the heterogeneity of the TGW population according to demographic and socioeconomic conditions. Initially, six seeds were selected and each one received three invitations to recruit three other TrTW from their network. Moreover, each new guest who participated in the study received three more invitations setting recruitment waves until reaching the "N" sample. Concerning seed randomness, we should point out that the greater the number of waves formed by a seed, the lower the degree of relationship in the last waves.

The RDS method includes an incentive/compensation scheme for the time dedicated to research and travel to the data collection site. Thus, each participant received BRL 30.00 (equivalent to 10 USS at the time of the study) for their participation, plus a secondary incentive of BRL 30.00 for each participant that they recruited and who participated in the research. Moreover, all received educational material, snacks, water-based lubricating gel, condoms, and a beauty kit (a necessaire-type bag containing lipstick, nail polish, and a mirror). The participants were recruited from September 2014 to April 2016 in a space organized for this purpose, located in Salvador, with opening hours from 1:00 pm to 5:00 pm, Monday through to Friday. Data was collected through interviews with an electronic, standardized, and tested questionnaire carried out by trained interviewers.

### 2.2. Study Variables

The outcome variable is the use of industrial liquid silicone (yes and no). The ILS use variables were: the age at which ILS was used for the first time ($\geq$25 years or <25 years); lifetime use of ILS (only once or 2–10 times); who applied the ILS (pumped ladies or others); how much ILS was applied to the body (<10 cups or >10 cups); and guidance received on ILS use (health professional, pumped ladies, and other travesti). Other study variables were: age (>25 years vs. $\geq$25 years); schooling ($\leq$8 years of study vs. >8 years); non-white (categorized by the combination of black, brown) vs. white skin color; occupation (formal work/informal work/businessperson vs. unemployed); income (Brazilian minimum wage per month in Brazilian reais ($\geq$1000 R$ vs. <1000 R$); gender identity (travesti (yes, no) vs. transgender women); marital status (living with a companion vs. singles); social name on the Brazilian National Health System (SUS) card (no vs. yes); use of female hormones (no vs. yes); satisfaction with body image (no vs. yes); feeling pleasure with the penis (no vs. yes); being comfortable with the penis (no vs. yes); experience of discrimination in health services (no vs. yes); illicit drug use in a lifetime (no vs. yes); sex work history (no vs. yes); condom use during insertive anal sex with clients (always use vs. inconsistent use); condom use during receptive anal sex with clients (always use vs. inconsistent use); and HIV testing before the study (yes vs. no). All participants received pre-test and post-test counseling and guidance before and after receiving the results of their tests. A whole blood rapid test by finger prick was performed for HIV. If this rapid test was reactive, a second confirmatory one with a different brand was performed following the HIV testing algorithm of the Brazilian Ministry of Health [16]. In addition, a whole blood rapid test was performed by finger prick for syphilis. Individuals with the first non-reactive test were classified in this study as uninfected. Individuals with both screening and confirmatory positive tests were classified as infected, also following the testing algorithm of the Brazilian Ministry of Health. All participants considered to have had a positive rapid test result were referred to public health services for confirmatory tests, monitoring, and treatment of the infection.

### 2.3. Data Analysis

Data analysis took into consideration the complex sampling design of the recruitment by RDS methodology, i.e., the dependence between observations resulting from referral chains and the probabilities of unequal selections due to the different sizes of each of the participant's network. The questions that measured the size of the social network of each TrTW's network of contacts were: "How many travesti/transgender women do you know by name and who also know you by your name in Salvador?" and "Of the travesti and

transgender women you know, how many would you invite to participate in this survey?". The sample was weighted by the RDS-II estimator [17]. The analysis was conducted using STATA version 14 (www.isc.ufba.br, 2015 (accessed on 5 May 2022)).

Descriptive analyses were performed to characterize the profile of the population and the variables related to ILS use, namely, bivariate analysis with an estimated prevalence of ILS use by study variable and Pearson's chi-square test considering a statistical significance level of 20% for variable selection for multivariate analysis. Adjusted prevalence ratio estimates and respective 95% confidence intervals (*p*-value of <0.05) were obtained using Poisson regression.

*2.4. Ethical Aspects*

All subjects gave their informed consent for inclusion before they participated in the study. The study was conducted in accordance with the Declaration of Helsinki, and the protocol was approved by the Ethics Committee of the Health Secretariat of the State of Bahia who approved the PopTrans study under N° 225.943 and CAAE 07135912.7.0000.0052, meeting all of the requirements defined in CNS Resolution N° 466/2012. Thus, the autonomy and dignity of the participants were respected, ensuring their willingness to remain, or not, in the research, and that all foreseeable harm would be avoided. Participants signed the Informed Consent Form or the Informed Assent Form when under 18 years of age.

**3. Results**

Table 1 shows the sociodemographic and behavioral variables and the prevalence of sexually transmitted infections. Among the 127 participants, 57.2% were under 25 years old, 51.6% were self-declared black or mixed race, and 48.4% were white. As for schooling, most (73.2%) reported above eight years of study.

The TrTW who were unemployed or engaged in sex work accounted for 71.3% (n = 88); the others were formally or informally employed 28.7% (n = 39); 51% reported income ≥R$ 1000.00; most (63.5%) were single at the time of the survey. As for gender identity, 31.5% self-identified as travesti and 47.9% as transgender women.

Regarding body modification, most (94.8%) reported the use of female hormones; 52.8% (n = 61) stated that they were not satisfied with their body image; 75.2% (n = 93) felt comfortable with their penis; and 69.3% (n = 87) reported feeling pleasure with their penis. Regarding sex work history, 77.6% (n = 90) engaged in sex work; 64.7% (n = 75) and 60.8% (n = 77) reported always using condoms in insertive anal sex and receptive anal sex with their clients, respectively; and 64.4% (n = 65) reported using illicit drugs at least once in their lifetime. A total of 48.4% reported being discriminated against in health services; 80.9% reported that they did not have a SUS card with their social name; 9% tested positive for HIV; and 31.6% for syphilis.

Table 2 describes ILS use variables among 44 TrTW. The weighted prevalence of ILS use was 31.6%. Regarding age, 70.6% injected ILS under 25 years of age; 49.3% used it only once; and 50.7% injected it between 2–10 times. ILS was primarily applied by pumped ladies (83.7%). As for the amount of injected silicone, 85.9% reported that they had applied >10 cups. A total of 79.7% stated that they received some guidance on ILS use through pumped ladies or other travesti, and only 20.3% obtained information through a health professional.

The prevalence of ILS use according to study variables is described in Table 3. White TrTW (61.8%); those who were unemployed (82.3%); and those who had an income above BRL 1000.00 (67.1%) used ILS more (*p* < 0.20). A statistically significant relationship with the use of ILS at *p*-value < 0.05 was observed for those who did not have a social name on the SUS card (67.4%) (*p* = 0.04) more than those who felt comfortable with their body image (75.5%); those who had a sex work history (92.3%); those who used illicit drugs in a lifetime (83.3%); and those that performed HIV testing before the study (95.1%).

**Table 1.** Description of the study population in Salvador, Northeast Brazil., Brazil, 2016.

| Variables | N | % | % * |
|---|---|---|---|
| **Sociodemographic and cultural** | | | |
| **Age** | | | |
| ≥25 years | 60 | 47.2 | 42.8 |
| <25 years | 67 | 52.8 | 57.2 |
| **Schooling years** | | | |
| >8 years | 79 | 61.4 | 73.2 |
| Up to 8 years | 48 | 31.6 | 23.8 |
| **Ethnicity/skin color** | | | |
| White | 63 | 49.6 | 48.4 |
| Black/brown | 64 | 50.4 | 51.6 |
| **Occupation** | | | |
| Formal work/informal work/businessperson | 39 | 30.7 | 28.7 |
| Unemployed | 88 | 69.3 | 71.3 |
| **Income** | | | |
| ≥1000 R$ | 71 | 55.9 | 51 |
| <1000 R$ | 56 | 44.1 | 49 |
| **Marital status** | | | |
| Living with companion | 35 | 27.6 | 36.5 |
| Single | 92 | 72.4 | 63.5 |
| **Gender identity** | | | |
| Travesti | 60 | 47.2 | 31.5 |
| Transgender woman | 67 | 52.8 | 47.9 |
| **Social name on SUS card** | | | |
| No | 103 | 81.1 | 80.9 |
| Yes | 24 | 18.9 | 19.1 |
| **Body change and pleasure with penis** | | | |
| **Using hormones or former hormone user** | | | |
| No | 9 | 7.1 | 5.2 |
| Yes | 118 | 92.9 | 94.8 |
| **Feeling comfortable with body image** | | | |
| No | 61 | 48.0 | 52.8 |
| Yes | 66 | 51.9 | 47.2 |
| **Feeling comfortable with penis** | | | |
| No | 34 | 26.8 | 24.8 |
| Yes | 93 | 73.2 | 75.2 |
| **Feeling pleasure with penis** | | | |
| No | 40 | 31.5 | 30.4 |
| Yes | 87 | 68.5 | 69.6 |
| **Sex work/condom use** | | | |
| **Sex work history** | | | |
| No | 37 | 29.1 | 22.4 |
| Yes | 90 | 70.9 | 77.6 |
| **Condom use during insertive anal sex with clients** | | | |
| Inconsistent | 36 | 32.4 | 35.3 |
| Always | 75 | 67.6 | 64.7 |
| **Condom use during receptive anal sex with clients** | | | |
| Inconsistent | 34 | 30.6 | 39.2 |
| Always | 77 | 69.4 | 60.8 |
| **Discrimination and drug use history** | | | |
| **Discrimination history in health services** | | | |
| No | 63 | 59.4 | 51.6 |
| Yes | 43 | 40.6 | 48.4 |
| **Use of illicit drugs in a lifetime** | | | |
| No | 62 | 48.8 | 35.6 |
| Yes | 65 | 51.2 | 64.4 |

**Table 1.** *Cont.*

| Variables | N | % | % * |
|:---:|:---:|:---:|:---:|
| **Sexually transmitted infections** | | | |
| **HIV testing before the study** | | | |
| Yes | 90 | 70.8 | 78.9 |
| No | 37 | 29.1 | 21.3 |
| **HIV test** | | | |
| Negative | 112 | 88.2 | 91.0 |
| Positive | 15 | 11.8 | 9.0 |
| **Syphilis test** | | | |
| Negative | 79 | 62.2 | 68.4 |
| Positive | 48 | 37.8 | 31.6 |

* Analysis weighed by RDS estimators.

**Table 2.** Use of industrial liquid silicone among travesti and transgender women. Salvador, Northeast Brazil, 2016 (n = 44).

| Variables | N | % | % * |
|:---:|:---:|:---:|:---:|
| **Use of industrial liquid silicone** | | | |
| No | 83 | 66.1 | 68.4 |
| Yes | 44 | 33.9 | 31.6 |
| **Age at first industrial liquid silicone use** | | | |
| <25 years | 32 | 72.7 | 70.6 |
| ≥25 years | 12 | 27.3 | 29.4 |
| **How many times have you used industrial liquid silicone in your life?** | | | |
| Only once | 22 | 50.0 | 49.3 |
| 2–10 times | 22 | 50.0 | 50.7 |
| **With whom did you apply industrial liquid silicone?** | | | |
| Pumped ladies | 41 | 93.2 | 83.7 |
| Others | 3 | 6.8 | 16.3 |
| **How many industrial liquid silicone cups have you applied?** | | | |
| <10 cups | 12 | 27.3 | 14.1 |
| >10 cups | 32 | 72.7 | 85.9 |
| **From whom did you receive guidance on the use of industrial liquid silicone?** | | | |
| Health professional | 4 | 13.8 | 20.3 |
| Pumped ladies or other travesti | 25 | 86.2 | 79.7 |

* Analysis weighed by RDS estimators.

In the multivariate analysis, the TrTW who felt more comfortable with their body image had a threefold higher prevalence of ILS use than those who were not comfortable with their body (PR = 3.01; 95% CI = 1.31–6.93). Those who performed an HIV test before the study had a 4.5 times higher prevalence of ILS use than those who did not (PR = 4.53; 95% CI = 1.29–15.89) (Table 4).

**Table 3.** Bivariate analysis of factors associated with the use of industrial liquid silicone among TrTW. Salvador, Northeast Brazil, 2016 (n = 127).

| Variables | Industrial Liquid Silicon Use | | |
|---|---|---|---|
| **Sociodemographic and Cultural** | Yes (%) * | No (%) * | *p*-Value |
| **Age** | | | |
| <25 years | 46.9 | 62.4 | |
| ≥25 years | 53.1 | 37.6 | 0.33 |
| **Schooling years** | | | |
| >8 years | 81.5 | 68.9 | |
| Up to 8 years | 18.5 | 31.0 | 0.23 |
| **Ethnicity/skin color** | | | |
| White | 61.8 | 41.6 | |
| Black/brown | 38.2 | 58.4 | 0.16 |
| **Occupation** | | | |
| Formal work/informal work/businessperson | 16.7 | 34.8 | |
| Unemployed | 83.2 | 65.2 | 0.13 |
| **Income** | | | |
| ≥1000 BRL | 67.1 | 42.7 | |
| <1000 BRL | 32.9 | 57.3 | 0.14 |
| **Marital status** | | | |
| Living with companion | 42.3 | 33.7 | |
| Single | 57.7 | 66.3 | 0.58 |
| **Gender identity** | | | |
| Transgender woman | 65.5 | 47.8 | |
| Travesti | 34.4 | 52.2 | 0.24 |
| **Social name on SUS card** | | | |
| No | 67.4 | 12.1 | |
| Yes | 32.6 | 87.9 | 0.04 |
| **Body change** | | | |
| **Using hormones or former hormone user** | | | |
| No | 4.2 | 5.8 | |
| Yes | 95.8 | 94.2 | 0.72 |
| **Feeling comfortable with body image** | | | |
| No | 25.5 | 66.9 | |
| Yes | 74.5 | 33.1 | 0.05 |
| **Feeling comfortable with penis** | | | |
| No | 24.2 | 25.1 | |
| Yes | 75.8 | 74.9 | 0.94 |
| **Feeling pleasure with penis** | | | |
| No | 31.8 | 29.7 | |
| Yes | 68.2 | 70.3 | 0.88 |
| **Sex work/condom use** | | | |
| **Sex work history** | | | |
| No | 7.7 | 30.0 | |
| Yes | 92.3 | 70.0 | 0.05 |
| **Condom use during insertive anal sex with clients** | | | |
| Inconsistent | 34.8 | 35.5 | |
| Always | 65.2 | 64.5 | 0.96 |
| **Condom use during receptive anal sex with clients** | | | |
| Inconsistent | 36.9 | 40.5 | |
| Always | 63.1 | 59.5 | 0.83 |
| **Discrimination and drug use history** | | | |
| **Discrimination history in health services** | | | |
| No | 68.3 | 70.1 | |
| Yes | 31.7 | 29.9 | 0.91 |
| **Use of illicit drugs in a lifetime** | | | |
| No | 16.7 | 45.3 | |
| Yes | 83.3 | 54.7 | 0.01 |

**Table 3.** *Cont.*

| Variables | Industrial Liquid Silicon Use | | |
| --- | --- | --- | --- |
| **Sociodemographic and Cultural** | **Yes (%) *** | **No (%) *** | ***p*-Value** |
| **Sexually transmitted infections** | | | |
| **Performed HIV testing before the study** | | | |
| No | 4.9 | 29.3 | |
| Yes | 95.1 | 70.7 | 0.003 |
| **HIV test** | | | |
| Negative | 88.1 | 92.5 | |
| Positive | 11.9 | 7.5 | 0.52 |
| **Syphilis test** | | | |
| Negative | 60.4 | 72.6 | |
| Positive | 39.6 | 27.4 | 0.38 |

* Analysis weighed by RDS estimators.

**Table 4.** Prevalence ratios of factors associated with industrial liquid silicon use in TrTW. Salvador, Northeast Brazil, 2016.

| Variables | Adjusted PR * | 95% CI |
| --- | --- | --- |
| **Schooling years** | | |
| Up to 8 years | 0.71 | 0.37–1.35 |
| >8 years | 1 | |
| **Ethnicity/skin color** | | |
| Black/brown | 1.01 | 0.57–1.79 |
| White | 1 | |
| **Occupation** | | |
| Unemployed | 0.98 | 0.44–2.16 |
| Formal work/informal work/businessperson | 1 | |
| **Income** | | |
| <1000 BRL | 0.80 | 0.39–1.63 |
| ≥1000 BRL | 1 | |
| **Gender identity** | | |
| Travesti | 0.75 | 0.40–1.41 |
| Transgender women | 1 | |
| **Social name on SUS card** | | |
| Yes | 1.08 | 0.66–1.76 |
| No | 1 | |
| **Feeling comfortable with body image** | | |
| Yes | 3.01 | 1.31–6.93 |
| No | 1 | |
| **Sex work history** | | |
| Yes | 1.81 | 0.58–5.58 |
| No | 1 | |
| **Use illicit drugs in a lifetime** | | |
| Yes | 1.66 | 0.82–3.89 |
| No | 1 | |
| **Performed HIV testing before the study** | | |
| Yes | 4.53 | 1.29–15.89 |
| No | 1 | |

* Analysis weighed by RDS estimators.

## 4. Discussion

The aim of this study was to describe the prevalence of ILS use and the estimated associated factors. The data presented here show that ILS was used by almost a third of the travesti and transgender women participating in the PopTrans study (31.6%). A cross-sectional study carried out by Pinto [18] showed a prevalence of ILS use of 49% among 576 travesti in the State of São Paulo. An RDS study carried out in San Francisco, California, estimated the use of ILS among 250 transgender women at 16.7% [19], while another cross-

sectional study carried out in Vietnam [10] with 205 transgender women noted that 39% of the participants had used ILS, and 59% of these had used it in inappropriate locations. These studies corroborate the illicit use of ILS in this population.

In Brazil, the ban on ILS use in health started with the then Representative Sebastião Rocha's bill (PL) in 1999. This bill was approved in 2005 when the use of liquid silicone became a bodily injury crime. As of 2008, there were no health policies for trans people in the Unified Health System (SUS), and GM/MS Ordinance N° 1707, which was revoked, defined the age of 18 for sex reassignment surgery [20]. Due to legal issues, in 2013, this ordinance was replaced by Ordinance GM/MS N° 2.803, which embraced, more broadly, the comprehensive care of trans people [21]. Besides the care of the transgenderizing process using silicone prostheses and sex reassignment surgeries, the monitoring of body modification with hormones was also implemented in outpatient clinics.

In our study, 70% of the TrTW were under 25 years of age when they used ILS for the first time, which may be related to the need to change the body quickly due to gender dysphoria, the low-volume effects of hormones in the breasts and glutes, and the need to value the body since most TrTW engage in sex work [22–24]. A cross-sectional study carried out in San Francisco, California [19], argues that ILS use at older ages may be associated with greater purchasing power and emphasizes that younger women more often support the side effects of these illicit fillers. In 2016, The Center of Excellence for Transgender Health published a guide to primary care and gender affirmation for trans and non-binary gender people [25]. These guidelines emphasize that the use of ILS, in general, is for rapid body change in order to alleviate gender dysphoria. In a qualitative study, [11] reports that the use of silicone by TrTW can be explained by low self-esteem, misperceptions about ILS, discomfort in public environments, and the need for feminization and being perceived as a woman (a process known as passability).

Another point is in epidemiological studies' difficulty in defining the ILS amounts applied to body regions by TrTW. The questionnaire attempted to use a language as close as possible to the daily lives of this population. Thus, the expression "silicone cups" was used in the questions. In an ethnographic study carried out in Salvador, Bahia, Brazil, Kulick [26] observed that TrTW above 17 years of age injected ILS, and the amount ranged from 18 L to a few cups of ILS. The author claims that each liter of silicone is equivalent to six "glasses of water". In our study, most TrTW used over ten silicone cups applied by pumped ladies, totaling around 2 L of injected ILS, which corroborates other ethnographic studies performed [6,27].

This study reported an association between sex work and the use of ILS, which may be related to the need to value the body amid sex work and the greater representation and approximation of this body to cisgender women, as mentioned in other cross-sectional studies with trans women [6,7,19,22,28]. Pelúcio [4] emphasizes that the dynamics of corporeality can take shape within the scope of prostitution. The author emphasizes that sex work can be understood differently by TrTW, such as generating income and, thus, creating a sociability environment. The conceptions of their bodies are closely linked to the search for beauty as a model for experiencing the female gender. Becoming feminine emphasizes a process of reframing rules in order to identify and mold oneself in a continuous process of learning and experimenting while experiencing another body [27]. In our study, the TrTW who felt more comfortable with their body image used ILS more frequently. As highlighted by Pelúcio [4], body feminization and transformation in TrTW are continuous and endless. It is more probable that the greater the ILS use, the greater the satisfaction with body self-image will be, and, as the body gains feminine contours and silhouettes, that can be more exuberant or not, the greater the reduction will be in the discomfort caused by gender dysphoria. Another critical point is the need for TrTW sex workers using ILS to increase their clientele in order to ensure payment for the application of silicone, which can make them more susceptible to some sexually transmitted infections [18].

We identified a higher prevalence of ILS use among those who tested for HIV before the study, which may have occurred due to the possibility of sharing needles and syringes

at "pumped ladies' parties", so that there may eventually be a "protection" or assessment of serology before sharing needles that inject silicone to avoid HIV and the possible transmission of other STIs. In these situations, as highlighted in the study by Pinheiro Junior et al. [29], a higher level of testing for HIV is found in risky behaviors such as sex without a condom and illicit drug use before or during sexual intercourse.

## 5. Conclusions

As citizens and civil rights holders, TrTW promote new dialogues in all spheres, whether in the construction of their bodies or the dynamics of life. On the other hand, as a constitutionally established health provider, the State must make efforts to advocate access to health services and the right to a decent life in vulnerable population groups such as travesti and transgender people in order to facilitate the broad implementation of this dialogue.

As discussed in this paper, while TrTW are aware of the possible health risks, the illicit use of ILS is ongoing due to the limited possibilities for body modification through the public health system. Furthermore, it is noteworthy that body change in trans people is part of a set of actions used in order to enable a new gender performance and, therefore, a right to a dignified and citizen-like existence. Unfortunately, this transformation is a time-consuming process through legal means due to waiting lists and a lack of training for SUS professionals.

### 5.1. Public Health in Brazil for Trans People

The right to access modification body technologies occur in the resolutions and ordinances referred to here. There are still access limitations due to few referral centers in Brazil. This results in a delay in service and, therefore, in the use of industrial liquid silicone for faster body modification. It is essential to highlight aspects of citizenship, rights, and human dignity in the process of receiving and caring for the health of trans people, considering their needs for body modification and body prostheses for self-recognition. It is also necessary to question the blaming of trans people for the disease processes and consider the problem as a matter of public and community health. Thus, we suggest extensive training in professional health care in order to refer this target audience. Despite attempts to advance, gender dissidence is still pathologized in the legal–medical scope, as emphasized by the Federal Council of Medicine here in Brazil.

### 5.2. Study Limitations

The RDS study has limitations, such as the study's estimates representing the social network recruited by the participants. Therefore, our estimates cannot be extrapolated to the TrTW population of Salvador. Another point to be addressed is that the small sample size decreased the power of statistically significant association at a $p$-value of 5%. The studied population is considered hidden, hindering access, even with recruiters and peers with the same characteristics. Despite these limitations, our study brings new possibilities to discuss public policies for the health of TrTW populations and new paths for future research.

**Author Contributions:** R.A.d.S.: Main author, contributed to the literature review, construction of tables, statistical analysis, and organization of the work for this research. I.D.: Supervisor of the PhD research work, revised the text and tables, and contributed to the theme. L.A.V.d.S.: Doctoral co-advisor, revised the text, and contributed to the theme. F.S.: Revised the text and contributed to the theme's content. All authors have read and agreed to the published version of the manuscript.

**Funding:** This research was founded by National Department of STD, AIDS and Viral Hepatitis/Ministry of Health.

**Institutional Review Board Statement:** The study was conducted in accordance with the Declaration of Helsinki, and approved by the Ethics Committee of Secretary of Health State of Bahia CAAE: 07135912.7.0000.0052 for studies involving humans.

**Informed Consent Statement:** Informed consent was obtained from all subjects involved in the study.

**Data Availability Statement:** The data in this study can be found with the authors or in Collective Health Institute, www.isc.ufba.br (accessed on 5 May 2022).

**Acknowledgments:** The authors wish to thank ATRAS/Millena Passos; IBCM/Father Alfredo Dorea; Keila Simpson; and Ailton Santos for encouraging the topic; the National Department of STD, AIDS and Viral Hepatitis/Ministry of Health. Project Approved by the Ethics and Research Committee of the Bahia State Health Department. Opinion 225.943, CAAE: 07135912.7.0000.0052.

**Conflicts of Interest:** The authors declare no conflict of interest.

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
