# Peer review of "Factors Associated with the Use of Industrial Liquid Silicone among Travesti and Transgender Women in Salvador, Northeast Brazil"

_venereology, doi:10.3390/venereology1030016_

Round 1

Author Response

Dear Reviewer

Thanks for your contributions. we Attached the answers.

Reviewer 2 Report

Factors associated with the use of industrial liquid silicone among travesti and transgender women in Salvador, Bahia, Brazil

This is an important contribution to the understanding of risk factors of TrTW in Salvador, Brazil. Nevertheless, I believe a few changes would improve the overall quality of the article.

1.     Including a paragraph or two regarding the social status of TrTw in Brazil, and the existence of stigma/transphobia around these identities would be helpful in the understanding of the use of the resources sought by these individuals.

2.     Line 68 – authors must provide further details when listing objectives (general and specific).

3.     The fact that the study included underaged participants poses legal/ethical questions. How were these issues handled?

4.     Line 89 – please provide an estimation of how much 30 reais is in USD or €.

5.     Data collection took place from 2014-2016. Authors should demonstrate if the data is still valid to the present day or if this is a retrospective study.

6.     Please provide more detailed information regarding questionnaire used, e.g., sociodemographic questionnaire, including response options.

7.     Travesti is a very “Brazilian” definition that doesn’t fit most western labels of gender identity. Please provide a comprehensive explanation of this phenomenon.

8.     Tables – please incorporate the * at the appropriate p-value in each column.

9.     Authors must include a limitations section.

10.  Authors must include an implications section, especially in terms of public health, but also in social inclusion policies in Brazil.

Best wishes.

Author Response

Dear Reviewer

Thanks for your contributions. Attached the answers

Reviewer 3 Report

The work presented is of enormous interest and relevance. The use of industrial liquid silicone among the transgender population is a constant that deserves attention from health professionals.

However, there are a number of aspects that the authors should review in the submitted manuscript:

- It would be advisable for the authors to specify in a more detailed way what they mean by "transvestites and transgender women".

- In line 54 - 55 there is a lack of references to support this statement.

- The authors note something about the context in which these practices take place in the discussion. For a better understanding of what is happening, it would be interesting if this were explained more in the introduction.

- The bibliographical references used are, in general, old. It would be important to update the references.

- The sample was collected 8 years ago, why did the authors not publish this data earlier?

- It would be necessary to include at the end of the discussion a section on the limitations of the study and another on possible future lines of research.

Author Response

Dear Reviewer

Thanks for your contribution. Attached the answers

Round 2

Reviewer 1 Report

Line 367

Please put the "study limitations" section after the "Discussion" section and before "Conclusions"

Line 467 and 470

"BRAZIL"

Remove the capital letter

Line 467

The correct link is this: https://bvsms.saude.gov.br/bvs/saudelegis/gm/2013/prt2803_19_11_2013.html

Reviewer 2 Report

Thank you for addressing almost all the requested changes; I believe the article is now fit for publication.

Best wishes.

Reviewer 3 Report

The changes made are in accordance with the requirements made by this reviewer.